# Global crop yields can be lifted by timely adaptation of growing periods to climate change

Sara Minoli ●[1] ✉, Jonas Jägermeyr[1,2,3], Senthold Asseng ●[4], Anton Urfels[5,6,7] & Christoph Müller ●[1]

Adaptive management of crop growing periods by adjusting sowing dates and cultivars is one of the central aspects of crop production systems, tightly connected to local climate. However, it is so far underrepresented in crop-model based assessments of yields under climate change. In this study, we integrate models of farmers' decision making with biophysical crop modeling at the global scale to simulate crop calendars adaptation and its effect on crop yields of maize, rice, sorghum, soybean and wheat. We simulate crop growing periods and yields (1986-2099) under counterfactual management scenarios assuming no adaptation, timely adaptation or delayed adaptation of sowing dates and cultivars. We then compare the counterfactual growing periods and corresponding yields at the end of the century (2080-2099). We find that (i) with adaptation, temperature-driven sowing dates (typical at latitudes >30°N-S) will have larger shifts than precipitation-driven sowing dates (at latitudes <30°N-S); (ii) later-maturing cultivars will be needed, particularly at higher latitudes; (iii) timely adaptation of growing periods would increase actual crop yields by ~12%, reducing climate change negative impacts and enhancing the positive $CO_2$ fertilization effect. Despite remaining uncertainties, crop growing periods adaptation require consideration in climate change impact assessments.

Plant phenology is the sequence of morphological and physiological events in the annual plant growth cycle. As the onset and rate of progress of phenological events are primarily driven by temperature and day length, the rising global temperatures due to climate change alter the timing of plants' phenology. Phenological records of the recent past in Europe show that the responses of plants to temperature increase have determined earlier and faster phenological progress of both wild and cultivated plants[1,2]. Faster-growing cycles associated with higher temperatures are considered one of the main mechanisms of climate change impacts on crop yields[3,4].

However, the phenology of annual crops depends also on farmers' decisions on sowing dates and varietal choice[5–7]. Annual crops have growing-period durations (here defined as days from sowing to maturity) of less than a year. They are thus exposed only to a fraction of the annual weather[8], and farmers manage crops to enhance yields and resource use by growing them during the most favorable part of the

[1]Potsdam Institute for Climate Impact Research (PIK), Member of the Leibniz Association, 14412 Potsdam, Germany. [2]NASA Goddard Institute for Space Studies, New York, NY 10025, USA. [3]Columbia University, Climate School, New York, NY 10025, USA. [4]Technical University of Munich, Department of Life Science Engineering, 85354 Freising, Germany. [5]Sustainable Intensification Program, International Maize and Wheat Improvement Centre (CIMMYT), South Asia Regional Office, Khumaltar, Lalitpur 44700, Nepal. [6]Water Resources Management Group, Wageningen University & Research, 6708 PB Wageningen, the Netherlands. [7]Centre for Crop Systems Analysis, Wageningen University & Research, 6708 PB Wageningen, the Netherlands. ✉e-mail: sara.minoli@pik-potsdam.de

year. Through plant domestication and breeding, humans have artificially modified the response of cultivated varieties (cultivars) to temperature and day length and by that, have expanded the area where crop species can be grown[8,9]. Farmers can thus draw from a vast assortment of cultivars that differ by their thermal time (maturity classes), photoperiod (long-day, short-day or day-neutral) and vernalization requirements (vernalizing, non-vernalizing or facultative), which are considered central for adapting cropping systems to changing climatic conditions[9].

It is essential to accurately represent growing periods in crop simulation models for estimating yield responses to climatic factors[10]. At the global scale, calibration of crop growing periods to reported historical crop calendars[11–13] has been shown to improve simulation of observed interannual yield variability[14,15]. However, for climate impact assessments, previous studies have not assessed farmers' adaptive behavior and relied on observation-based static crop calendars, thereby either assuming unchanged sowing dates and cultivar selection[13,16], or simply adapting crop cultivars to regain the duration of the reference growing period that is otherwise shortened with accelerated phenology under climate change[17].

Studying changes in global crop growing periods is constrained by the lack of sufficient information on management (e.g., sowing and harvest dates, cultivar choice), the timing of phenological phases (e.g., flowering, maturity), and crop development parameters (e.g., growing degree days, base temperatures, sensitivity to photoperiod). Recent efforts have started to explore decision-making rules that dynamically simulate farming practices to adjust crop growing periods and have found that climate is the primary driver of farmers' sowing date and cultivar choices[18–22], explaining the current patterns of cropping calendars. Yet, the adaptation of such practices under climate change and their impact on future crop yields remains poorly understood at the global scale.

Here, we investigate how accounting for farmers' adaptive management affects estimates of future global crop yields under climate change. We simulate sowing and maturity dates adaptation by combining two rule-based methods[19,22]. These methods simulate farmers' crop-calendar selection worldwide based on agro-climatic principles that are primarily driven by crop physiology and long-term changes in monthly temperature, precipitation and their seasonality. We improve these methods and jointly evaluate them against observed crop calendars[11–13] by driving them with observation-based climate[23]. Thereafter, we derive projections of location-specific sowing and maturity dates adapted to one historical (1986–2005) and two future (2060–2079, 2080–2099) climatic periods for five crops (maize, rice, sorghum, soybean, wheat) and two water management regimes (rainfed and irrigated). To represent cultivars adapted to these crop calendars, we compute thermal unit requirements (TU$_{req}$, °C day) between sowing and maturity dates[15] in each period.

We use the process-based global gridded crop model LPJmL[24,25] to simulate annual crop growing periods and yields from 1986 to 2099 under a reference scenario of the actual crop- and grid-cell-specific historical management practices (irrigation, fertilization, tillage, and crop residue management), assuming that no management action would be taken in response to climate-induced impacts. We then compare the counterfactual growing periods (with adaptation) and corresponding yields at the end of the century (2080–2099) in order to quantify how much (i) future sowing and maturity dates would shift under different adaptation and climate scenarios and (ii) future crop yields could be enhanced, if farmers adapted growing periods to climate change. We simulate three counterfactual sowing date and cultivar scenarios where farmers are assumed to (i) continue apply all historical management practices, including sowing dates and cultivars, also in the future (*no adaptation*); (ii) timely adapt sowing dates and cultivars to future climate (*timely adaptation*); (iii) adapt sowing dates and cultivars with 20-years delay (*delayed adaptation*) (Table 1).

**Table 1 | Growing-periods adaptation protocol**

| Simulation setup | Period of growing-period calibration | Period of simulation |
|---|---|---|
| Reference | 1986–2005 | 1986–2005 |
| No adaptation | 1986–2005 | 2080–2099 |
| Delayed adaptation | 2060–2079 | 2080–2099 |
| Timely adaptation | 2080–2099 | 2080–2099 |

For each simulation setup, the time interval of growing-periods calibration and crop-model simulation are specified. Sowing dates, maturity dates, and resulting cultivar thermal unit requirements (TUreq) are derived in each grid cell under the climate conditions of the respective period. Growing-period calibrations and crop-model simulations for each period are performed for four GCMs (HadGEM2-ES, GFDL-ESM2M, IPSL-CM5A-LR, MIROC5), respectively. See "Methods" for more details.

We also test the sensitivity of yields to single measures, adapting either sowing dates or cultivars individually, while keeping the other fixed at the reference level (*sowing-day only* and *cultivar only* adaptation).

In each simulation unit, different crops are assumed to grow on separate stands and to have only one crop cycle (sowing-to-harvest) per year. Therefore, multi-cropping systems (e.g., rice-rice systems in South-East Asia) are not explicitly represented. Adaptation measures need to be climate-scenario specific and we test several General Circulation Models (GCMs) to assess how robust results are against uncertainties in climate projections (Supplementary Figs. 1–4). Further details are described in "Methods".

## Results
### Simulated present-day and future adapted sowing dates
At latitudes higher than roughly 30°N–S, growing periods are mostly temperature-driven (Supplementary Fig. 5), with sowing dates that depend on the start of either the warm (for spring crop types), or the cold (for winter crop types) season. In the tropics, growing periods are largely driven by precipitation seasonality (Supplementary Fig. 5), with sowing dates occurring at the onset of the main rainy season (first of the 120 consecutive wettest days in the year), except for winter wheat that depends on temperature seasonality only, if present. For wheat, we improved the definition of winter and spring types compared to the original approach[19,22], including the following combinations of sowing seasons and vernalization requirements of the cultivars: *winter wheat with vernalization*, sown in fall and going dormant over winter; *winter wheat without vernalization*, sown in fall and grown over mild winters; *spring wheat without vernalization*, sown in spring. This distinction allows to better simulate winter wheat growing seasons in warmer climates, where temperatures are not low enough for dormancy (e.g., India) (Supplementary Fig. 6).

*Timely adaptation* to future climate (2080–2099) markedly shifts temperature-driven sowing dates in the extratropics (latitudes > 30°N–S). Due to an earlier spring onset, spring crops (maize, rice, sorghum and soybean) show advancing sowing dates between −30 and −10 days, depending on the region and GCM (Fig. 1a and Supplementary Figs. 7–10A). An earlier spring allows for taking advantage of a longer growing season. In contrast, adapted sowing dates of *winter wheat with vernalization* are postponed (+10 to +30 days) (Fig. 1a and Supplementary Figs. 7–10A), to avoid that the crop develops too much before going dormant, which would make it more susceptible to frost damages and disease. Larger differences (>+60 days) found for wheat in high latitudes (>50°N–S) reflect the switch from spring to winter types, meaning that sowing dates are pushed several months forward, from spring to fall. This is the case in regions across North America, Europe, and Asia, which in the 1986–2005 reference period the rule-based model finds to be only suitable for growing spring wheat because of too long and harsh winters, become suitable for growing winter wheat under climate change (shorter and warmer winters).

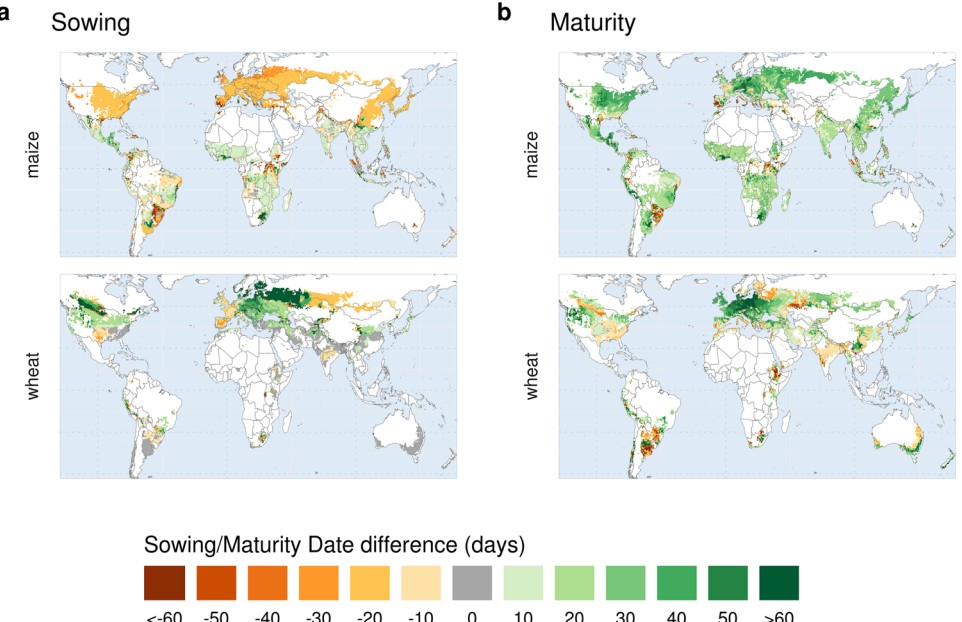

**Fig. 1 | Crop-calendars adaptation.** Differences (days) in simulated average sowing (**a**) and maturity (**b**) dates between *timely adaptation* and *no adaptation* scenarios for the same climate period (2080–2099, RCP6.0). Crop calendars are climate-scenario (GCM) specific, here the mean changes across four GCMs are shown. Results for individual GCMs and all crops are reported in the Supplementary Information.

Sowing dates of *winter wheat without vernalization* remain mostly unchanged or even advanced, as no direct frost damage is expected over mild winters (without considering possible spring frost damage at anthesis). Other factors, i.e., diseases, might affect sowing date decisions, which however, are not considered in our purely climate-driven approach. With few exceptions, precipitation-driven sowing dates adapted to future climate show little changes (±10 days) across all GCMs compared to the reference scenario. This indicates that, despite the substantial changes in rainfall (Supplementary Fig. 4), the average timing of the onset of the main rainy season is projected to not change much in these particular climate scenarios (Fig. 1a and Supplementary Figs. 7–10A). Yet, future approaches would need to address also changes in climate and growing periods of interannual variability.

## Future maturity dates and cultivars adaptation

Changes between future adapted and non-adapted maturity dates reflect two different processes. On the one hand, non-adapted crops are subject to an overall shortening of the growing-period durations compared to the historical ones (Supplementary Fig. 11). This is a consequence of increased temperatures, which drive faster phenological development. On the other hand, the adaptation rule seeks the most suitable maturity dates adapted to future climate, aiming to extend the growing period while avoiding hot and/or dry periods that could hamper grain filling and reduce final yields. For example, maturity dates of spring crops occur at the end of the summer, so that the reproductive phase (grain filling) takes place after temperatures have reached their seasonal peak. Specifically, if temperatures remain within an optimal range throughout the year (Supplementary Table 1), grain filling starts in the warmest month of the year. If instead, temperatures exceed crop-specific optima, longer growing periods are preferred, to place grain filling after the hottest period has passed. Therefore, *timely adapted* maturity dates are generally later than with *no adaptation* (Fig. 1B and Supplementary Figs. 7–10B). Yet, earlier maturity dates are found in regions with water limitations, where maturity is anticipated to avoid terminal water stress.

Winter wheat, on the other hand, matures in the warmest month of the year, if monthly average temperatures remain below 25 °C (see Supplementary Table 1). However, if temperatures exceed 25 °C, wheat matures on average on the day this threshold is exceeded, in order to escape terminal heat stress (note that daily temperatures are derived by linear interpolation of monthly values, see "Methods"). As a result, we find that *timely adaptation* at mid-to-high latitudes, where temperatures remain within an optimal range under future climate, leads to later wheat maturity. In contrast, ideal maturity dates are advanced across mid-latitude regions where monthly temperatures will exceed 25 °C. This acts in the same direction as the phenological shortening of the growing period that occurs under the non-adapted scenario (Supplementary Fig. 11), resulting in small maturity dates differences (<10 days) between the two scenarios (Fig. 1b and Supplementary Figs. 7–10B). At very high latitudes (e.g., Canada, Russia), where a switch from spring to winter wheat is expected to occur, maturity dates are also advanced, as wheat sown in fall generally matures earlier.

Cultivars adapted to future climate generally require accumulating more thermal units to reach maturity than the non-adapted cultivars. The largest shifts are found in the extratropics, where the growing seasons extend in both directions, with earlier sowing and later maturity (except for wheat) (Supplementary Fig. 12). Tropical regions, do not indicate many variations in sowing dates, but adaptation of later-maturing cultivars allow for regaining some of the growing-period length that is "lost" due to accelerated phenology (Supplementary Fig. 11).

## Yield responses to crop-calendar adaptation

Global crop yields on current cropland, simulated by the process-based model LPJmL, benefits from phenological adaptation under future climate (RCP6.0; 2080–2099). For all crops combined, global yields would be 12% (GCM range 9–15%) higher compared to a scenario without adaptation (Fig. 2). The largest yield gains are found for maize (+17%) and rice (+17%), and the smallest for wheat (+7%). The benefits of adaptation reflect the warming level of the climate scenario, being generally larger under the warmer (HadGEM2-ES) than under the cooler (GFDL-ESM2M) climate scenario[26] with some variations across crops, due to different spatial patterns of warming (Supplementary Fig. 3) and cultivation area (Supplementary Fig. 13).

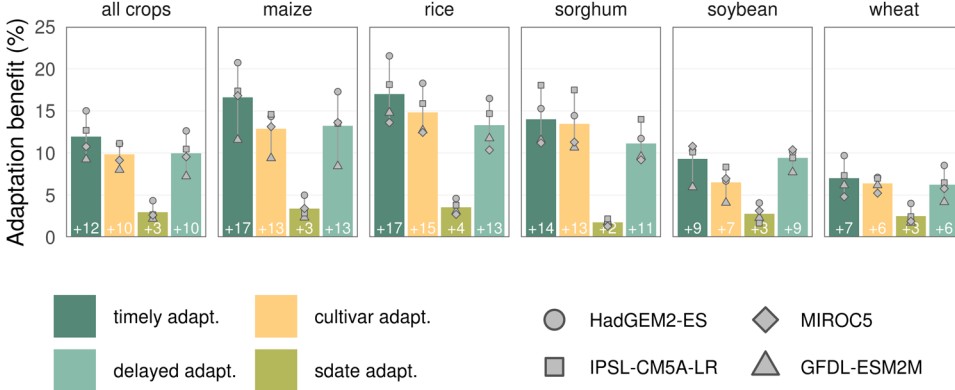

**Fig. 2 | Benefits of sowing and cultivar adaptation on global crop yields (2080–2099, RCP6.0).** Benefits on global yields are reported for all crops aggregated and for each individual crop, along with the uncertainty under different climate scenarios. The four adaptation scenarios indicate different levels of adaptation (adapt.): *timely adaptation*, sowing dates and cultivars adapted as the climate is changing (2080−2099); *cultivar adaptation*, sowing fixed at the reference level, only cultivars adapted as in timely adaptation; *sowing date adaptation*, only sowing dates adapted as in timely adaptation, cultivars fixed at the reference level; *delayed adaptation* both sowing dates and cultivar adapted but with 20-years delay, to 2060−2079 climate. The global yield of an individual crop is computed as the area-weighted mean yield across all grid cells growing that crop. In grid cells where adaptation of growing periods returned either no benefit or maladaptation (yield difference is equal or larger zero) yield losses were considered equal zero. Bars represent the mean across GCMs ($n = 4$ GCMs), whiskers display the range across GCMs, and gray symbols refer to individual GCMs.

Although adaptation is key for lifting yields of all considered crops under a warming climate, it is especially important for $C_4$ crops. Indeed, the increasing atmospheric $CO_2$ underlying the climate change scenarios considered here is able to over-compensate the negative impacts due to warming (−12%), leading to overall higher global crop yields at the end of the century (+14%) relative to the historical period, even without adaptation (Supplementary Fig. 14). As expected, this effect is more pronounced for $C_3$ crops (rice +24%, soybean +24% and wheat +22%) than for $C_4$ (maize −2% and sorghum 0%), which are far less responsive to $CO_2$ increase[27]. Furthermore, adaptation proved to be nearly insensitive to the assumptions on the level of atmospheric $CO_2$ in our simulations, as the responses to adaptation are similar under *elevated* (Fig. 2) and *static* $CO_2$ (Supplementary Fig. 15). This makes adaptation effects independent of projected $CO_2$ responses, which are still a large source of uncertainty in crop models[13,16]. Another interesting aspect is the combined effects of increasing atmospheric $CO_2$, climate change and adaptation on yield variability, which we quantify by the coefficient of variation (Supplementary Fig. 16). There is a general expectation that the interannual variability in crop yields will increase under climate change[28]. Our simulations support this thesis only under the (unrealistic) scenario without $CO_2$ increase (*no adaptation without $CO_2$*), whereas we find even slight declines in yield variability if $CO_2$ is included (*no adaptation*). For rice, sorghum and wheat, the two adaptation scenarios (*timely adaptation* and *delayed adaptation*) show a decreased variability compared to the *no adaptation* scenarios, because of negligible changes in standard deviation along with an increase in mean yield. On the other hand, yields of maize and soybean show increase in both mean and standard deviation, resulting in an overall higher coefficient of variation under adaptation scenarios, compared with *no adaptation*. However, also for these two crops neither the coefficient of variation nor the standard deviation of yields increase above those of the reference period.

The highest increase in yields is obtained when both sowing and cultivar adaptation are combined (except for wheat). Yet, the latter shows more important crop responses than the change in sowing dates (Fig. 2). This effect is explained by larger changes of maturity dates (Fig. 1), together with higher biomass accumulation rates (yield gain per day of growth) towards the end of the growing season. Moreover, we find that a *timely* adaptive response would be more effective than a *delayed* response (Fig. 2), which could be caused by e.g., breeding programs lagging behind rapidly changing climatic

conditions (*30*). Particularly, a timely adaptive response matters the most for maize, rice, and sorghum that show differences at the global level between *timely* and *delayed* adaptation scenarios of +4%, +4%, and +3%, respectively.

For each crop, a certain share of the cropland did not show an advantage for yields from adaptation (yields under *timely adaptation* ≤ yields under *no adaptation*). In these cases, as we only test one single adaptation measure here (changed growing periods), and we cannot distinguish if the negative effect is driven by inadequate assessment of future growing seasons or by the inability of LPJmL to respond to the adverse conditions that the growing season algorithm tries to avoid, we assume that farmers would rather continue applying historical crop management. Given that disadvantageous practices are unlikely to spread, we assume that, in such cases, farmers would stay with the only other option they have in our model setup that is to stay with the old growing period. Therefore, we do not consider the yield losses that would result from ineffective measures in those grid cells. The extent of this area is about 8% (7−9% across GCMs) for maize, 7% (6−8%) for rice, 9% (4−13%) for sorghum, 17% (9−31%) for soybean, 42% (38−45%) for wheat.

### Benefits from growing-periods adaptation vary by region and crop

The benefit of adapting sowing date and maturity type to climate change varies across regions and crops (Fig. 3). Largest yield gains (>30%) are most pronounced in high-latitude temperate regions, where low temperatures are currently a limiting factor to crop growth. Warming produces the dual effect of stimulating crop growth (e.g., temperatures closer to optimum for photosynthesis) and of extending the time window of favorable conditions for crop growth. Adaptation of growing periods to the longer favorable season is particularly suitable in these regions. Climate change might thus help expand crop yields if sowing and cultivars are adapted simultaneously.

Adaptation has considerable effects (10−30%) throughout the tropics, as well as in many other main breadbasket regions, including the Corn Belt in the USA (maize and soybean), South-East Asia (rainfed rice), the Sahel in Africa (sorghum), the North China Plains (irrigated rice), where these crops indicate important potential benefits from adaptation (Fig. 3b−e).

The overall small effectiveness of adaptation on total crop yields across mid-latitude regions (Fig. 3a) is caused by largely ineffective

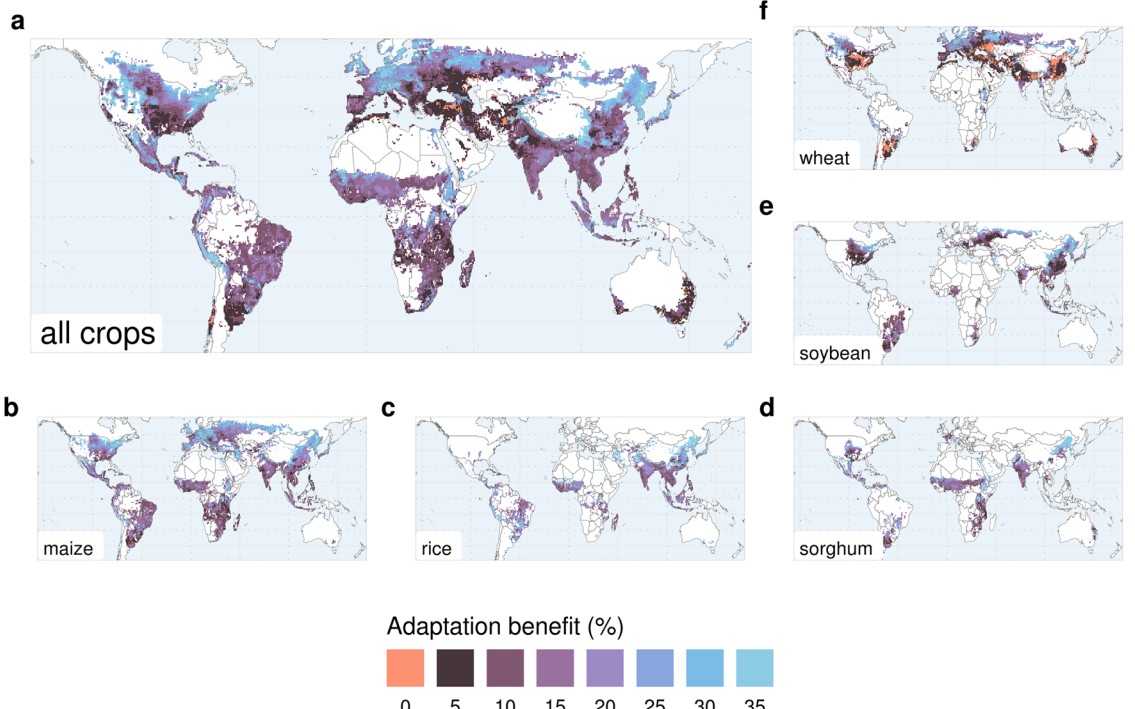

**Fig. 3 | Geographic patterns of crop-yield benefits from sowing and cultivar adaptation.** Yield benefits are computed as the relative difference (%) between the *timely adaptation* and *no adaptation* scenarios in the same climate period (2080–2099, RCP6.0). No adaptation indicates a scenario in which crop sowing dates and cultivars remain unchanged compared to the reference period (1986–2005). Yield benefits are reported for (**a**) all five crops aggregated (area-weighted mean of yields per grid cell) and for (**b**–**f**) each individual crop (maize, rice, sorghum, soybean and wheat). Adaptation benefits are shown as the mean across four GCMs. Maps for individual GCMs are reported in the Supplementary Information. In grid cells where adaptation lead to either no benefit or maladaptation (negative yield change), adaptation is not considered.

adaptation of wheat, a crop that comprises a large part of total production in these areas (e.g., central USA, Mediterranean, Northern India, and Central Asia) (Supplementary Fig. 13). The pattern is consistent across GCMs (Supplementary Figs. 17–20), and it could indicate that either the crop-calendar adaptation rules cannot always find better suitable growing periods or that the crop model LPJmL is insensitive to the adverse weather conditions that the adaptation rule tries to avoid. The latter is more plausible as, while the crop-calendar model rules for sowing and maturity dates consider terminal heat stress or over-winter crop damages and tries to select a shorter season in order to avoid them, the LPJmL crop model misses adequate representation of these important processes. Further efforts are required to better assess winter-wheat adaptation in these regions.

## Discussion

Current farming practices such as sowing date and cultivar choices are the result of gradual adjustments and optimization of the cropping systems to local conditions and incremental climatic change[29–32]. When adaptation in management is not considered, climate change impacts on crops might be overestimated, as previously shown by regional studies. For instance, temperature increases have been found to explain only half of the observed negative trend in heading date (timing of inflorescence emergence) of winter wheat in Germany, while recent faster-maturing cultivars account for the rest[33]. Similar adaptations have been reported for winter wheat in China[34]. In the USA, maize yields have increased despite temperature increases, partially owing to earlier sowing and longer-season cultivars[35]. Experimental trials show that improved wheat cultivars with delayed anthesis and increased grain-filling rate[36], or late-maturing cultivars with early sowing[37], can increase yields under the currently warming climate. At the global scale, the scarcity of information on crop management and its temporal evolution challenges the implementation of adaptation

scenarios in crop models[28], so that previous assessments had to rely on observed crop calendars[17].

Here, we present an agro-climatic rule-based approach to project farmers' decision-making rules on sowing dates and cultivar selection into future scenarios and to assess the resulting effects on yields of five major staple crops on current global cropland. Typically, regional and local studies find similar mechanisms of growing-period adaptation. In Europe, temperature-driven sowing of maize has been projected to occur earlier under future climate[6,30,38,39]. Similarly, the tendency of farmers to select later-maturing varieties with warming climate to counteract the shortening of growing periods under increasing temperatures and to take advantage of a longer growing season (unless there is the risk of terminal water stress) has been shown for Europe, the USA, and China[30,34,35,40]. Yet, some adaptation decisions remain excluded from our approach, which does not consider neither multiple-cropping systems nor crop rotations. In these systems, the end of the first growing season often determines the beginning of the second, adding complexity that is not captured by our climate-only driven approach.

Climate change is expected to largely impact crop yields. The most recent global-scale estimates based on crop-model ensemble results (including LPJmL) show that, without accounting for adaptation under high-warming scenarios (RCP8.5), maize global yields might decline by −24%, wheat yields increase by +17%, whereas rice (+2%) and soybean (−2%) yields are only marginally affected[13] (Supplementary Table 3). Within that model ensemble, because of its strong response to atmospheric $CO_2$, LPJmL is on the more optimistic side, showing yield changes of −7%, +18%, +24%, +15% for maize, rice, soybean, and wheat, respectively, which are in line with those found in our study for the same crops (−2%, +24%, +24%, +22%) (Supplementary Fig. 14). Our finding that introducing adaptation might increase crop yields by 12%, demonstrates the importance of considering farmers' decisions in

global biophysical modeling under climate change. By conducting global-scale assessments accounting for locally (and climate-scenario specific) adapted growing seasons, which previously was only done for individual case studies (location or region-specific; e.g., refs. 38, 40), we are able to highlight the sizeable effect of growing-periods adaptation. In fact, adaptation not only avoids negative yield impacts on individual crops (e.g., maize), but also further enhances yields of those crops that benefit from climate change (e.g., wheat), supporting the necessary increase in food supply for a growing global population[41]. Similar to the effect of $CO_2$, adaptation should not be neglected in future crop yield projections. Climate change scenarios can now be supplemented with scenarios on crop growing season management in model experiments such as AgMIP[13,16] and ISIMIP[42]. This constitutes an important step in integrating dynamic management decision into global models of agricultural systems, representing a significant improvement over assuming static historical management practices in future climate change scenarios.

We here investigate farmers' decision-making that is only driven by their experience of climate conditions. Some of this adaptation will be implemented by the farmers directly (e.g., changing sowing dates[31]), as they will be observing changing conditions and will respond to these as good as they can. Bringing new cultivars into the field will, however, require concerted action of farmers, breeders, and markets, as suitable cultivars may not be directly transferable between regions,[18,43,44]. Whilst in regions with functional markets, breeders work to provide cultivars adapted to the farmers' needs[45], in others, the predominance of on-farm seed production systems can limit the introduction of new cultivars[46]. As the impact of climate change in such more vulnerable regions is largest, efforts are needed to exploit the full growing season adaptation potential, along with other measures to improve crop yields (e.g., improvement of soil fertility and water retention). These will depend on functional, fair, and sustainable markets in less developed regions, including the infrastructure necessary for accessing those markets[47].

We identify two limitations that LPJmL has in common with several other state-of-the-art global-scale crop models (e.g., ref. 48). First, the adaptation of sowing dates and cultivars will not only affect the start and end of growing periods, but also the timing of intermediate phenological phases, some of which are particularly susceptible to high temperatures (e.g., flowering). Although LPJmL simulates phenology through a single phase from sowing to maturity, anthesis could be identified as the time when allocation to storage organs starts. For a correct parameterization and evaluation of such additional phase and to project changes in flowering and grain-filling time under future climate, global datasets on crop flowering dates would be needed. Particularly, time series of flowering dates would also help the improvement of the phenology models, including the effect of photoperiod. The omission of photoperiod responses in the current and previous global modeling assessment studies, might have led to the oversensitivity of crop growing periods to increasing temperature (see ref. 49), which requires further investigation. Including intermediate phenological phases would also allow to test the sensitivity of the model to the use of phase-specific base temperatures[50], instead of a single base temperature as assumed by most global-scale models so far[17]. Second, LPJmL is a global crop model that is restricted in its representation of extreme-climate impacts on crop phenology and growth, such as heat stress, frost damage and soil water excess. Improving the representation of these impacts will be necessary to increase confidence in the estimated climate change impacts and potentials of growing periods adaptation on global food yields. For example, using air temperature instead of canopy temperature might introduce some biases in the simulation of crop phenological progress and other plant processes, particularly under water stress conditions. On the other hand, the modeling of canopy temperature is still subject to several challenges (e.g., less standardized measurements of canopy

temperature in the field to the data- and time-expensive calibration) and has not been tested globally[51], which makes it a longer-term objective for model development.

Finally, future research on adaptation potentials will have to advance on the integration of biophysical with socioeconomic aspects, e.g., land-use change and technological progress, that can have even larger effects than climate change and management[21,38,52,53]. It is plausible to assume that farmers and farming systems will adapt growing seasons to changing climate conditions. Here, we show how this can be accounted for in global assessments of climate change impacts on crop yields and that assessments not accounting for this could result being overly pessimistic.

## Methods

### Rule-based mean sowing and maturity dates
Location- and climate-specific mean crop calendars are computed by combining two rule-based approaches published by[19] and[22] to simulate sowing and physiological maturity dates of grain crops, respectively. The assumption is that farmers select growing seasons based on the mean climatic characteristics of their specific location and on the physiological limitations (base and optimum temperatures for reproductive growth; sensitivity to terminal water stress) of the respective crop species. Accordingly, they select sowing dates and cultivars with phenologies that, on average, meet these adapted maturity dates.

The climate is classified into (i) seasonality types, based on the coefficient of variation of monthly mean temperature and precipitation and (ii) temperature levels, based on the temperature of the warmest month as compared to the base and the optimum temperatures for the crop reproductive growth. Optimal temperatures for sowing, optimal temperature ranges for grain filling, as well as indicators of soil moisture conditions (based on precipitation/potential-evapotranspiration ratio (P/PET)), are defined as global parameters for each crop (Supplementary Table 1) and used as thresholds to identify the best timing for sowing and for the start or end of the crop grain-filling phase. To cope with fluctuations of daily values around these thresholds, mean daily temperature, precipitation and potential evapotranspiration are derived by linear interpolation between monthly values.

We distinguish between spring and winter crop types. Maize, rice, sorghum, and soybean are simulated as spring crops only, for wheat we simulate both types. For spring crops, farmers sow the crops at the onset of the wet season (first day of the wettest 120 consecutive days), in case of prevailing precipitation seasonality, or on the day of the year when temperatures increase above crop-specific temperature threshold[19] (Supplementary Table 1), in case of temperature-driven seasonality.

For wheat, we distinguish three types: *winter wheat with vernalization* is chosen if monthly temperatures fall below 0 °C, but winter is neither too harsh (temperature of the coldest month is higher than −10 °C), nor too long (temperatures fall below the sowing temperature threshold (12 °C) after 15th September (North hemisphere) or 31st March (South hemisphere)[19]. *Winter wheat without vernalization* is grown if winters are mild (the temperature of the coldest month is higher than 0 °C) without dormancy. In this case, wheat is sown 75 days before the coldest month of the year. This rule was arbitrarily chosen based on observed wheat sowing dates in mild winter regions. If the conditions for growing any of the winter-wheat types are not met (winter too harsh and too long), then *spring wheat* (without vernalization) is chosen. Note that the computed sowing dates do not differ between rainfed and irrigated for any of the crops.

The mean maturity date is chosen so that the crop grain-filling phase, the most critical for yield formation, occurs under the least stressful conditions possible in that location and climate as follows. Under precipitation seasonality, grain filling starts towards the end of the rainy season, when a P/PET threshold is crossed. Under temperature seasonality, (a) grain filling of spring crops starts in the warmest

month of the year (if summer temperatures are optimal), or right after temperatures return within an optimal range; (b) grain filling of winter crops ends in the warmest month of the year (if summer temperatures are optimal), or right before temperatures exceed the optimal range; (c) eventually, maturity is advanced to escape terminal water stress. Note that the grain-filling phase has a static duration of 60 days for maize and 40 days for all the other crops. This assumption is based on empirical relationships between the total growth period and the post-flowering reproductive phase, showing that the partition between the vegetative and reproductive phase of grain crops follows a saturation curve that levels off after 90–100 days of total growth duration[54]. Different crops are assumed to have only one crop cycle (sowing-to-maturity) per year, therefore neither multi-cropping systems nor crop rotations are accounted for in the decision-making rules. A detailed description of the rules and parameterization can be found in refs. 19, 22.

### Simulated crop calendars reflect current farmers' management

Simulated historical crop calendars, driven by the bias-corrected climate dataset WFDEI[23], largely agree with observations[11–13]. We compare results both at the country and grid-cell level because, although the observed crop calendars used here are gridded datasets, their underlying sources are often reported per country. The country-level comparison highlights that the agreement is good for most countries, importantly, including those with large cropland area. The area-weighted Mean Absolute Error (MAE) is close or well below 30 days for all considered crops (Fig. 4). The simulated crop calendars compare well with the observed data also at the grid-cell level. Large areas, including major agricultural regions of importance for global yields, show deviations within ±15 days for both sowing and maturity dates (Supplementary Table 2 and Supplementary Figs. 21–24). However, evaluating the accuracy below 30 days is limited by the time resolution of the observations, which is either (i) monthly[11] and converted by us into daily values, by taking the mid-day of the reported month, or (ii) daily[12,13], but resulting from averages over large time windows (often > 1 month). Overall, the accuracy of the model is in line with the original evaluations of this rule-base method[19,22], as well as with other studies simulating average growing periods across large regions[18,20].

### Simulation of daily crop phenology and yields with the LPJmL crop model

We perform a modeling experiment across the global land grid at 0.5° × 0.5° resolution. We used the LPJmL5 crop model[24,25] to simulate daily growth and phenological development of five crops, driven by climate projections from four General Circulation Models (GCMs) GFDL-ESM2M, HadGEM2-ES, IPSL-CM5A-LR and MIROC5 under the Representative Concentration Pathways 6.0 (RCP6.0) as provided in bias-adjusted form from the CMIP5 archive by the ISIMIP2b project[42]. Irrigated and rainfed production systems are simulated separately on their current harvested areas[11], which is also used to compute total crop yields at grid-cell and global scale, as the product of yield by crop-specific area. A first 5000-year spin-up simulation is used to initialize all model pools (e.g., soil carbon and nitrogen content). A second spin-up simulation of 390 years is used to introduced effects of historical human-driven land-use change on these pools. A change in cropping area for the future scenarios is not considered in this study.

Phenological development is simulated based on the thermal-time model, including the effect of vernalization. All crops are assumed to be insensitive to photoperiod, due to a lack of parameters for multiple-crops and global-scale simulations. Previous global studies[15,18] that have focused on maize and wheat only, have found lower performances in the growing-period simulations when using a photo-thermal model, compared to a temperature-only driven approach and thus recommend caution when using the photoperiodic response. State-of-the-art global crop models[13,16] also typically do not consider sensitivity

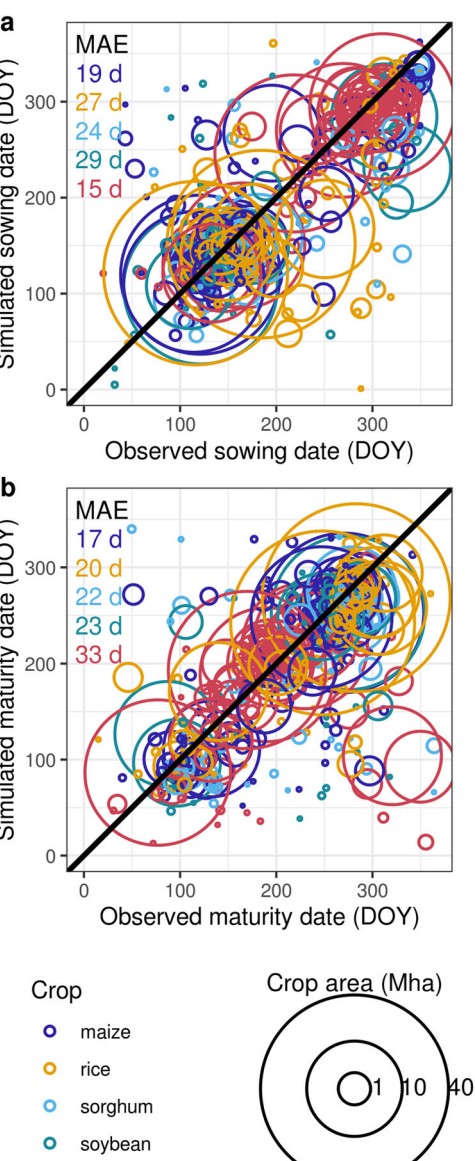

**Fig. 4 | Evaluation of simulated crop calendars.** Country-level comparison of simulated and observed sowing (**A**) and maturity (**B**) dates (day of the year) for five crops. Each circle refers to a country and a crop, the size of the circle is scaled according to the cropland area per country. The area-weighted Mean Absolute Error (MAE, days) is reported for each crop. Crop-calendar simulations are based on WFDEI reanalysis climate forcing[23] for the period 1979–2012. The observed crop calendar includes different sources[11–13].

to photoperiod or assume that the photoperiodic response of the cultivars chosen in each location are perfectly tuned to the given conditions.

Sowing dates are prescribed based on the external rule-based algorithm. Crop cultivars are parametrized based on the phenological units required to reach the corresponding maturity dates ($TU_{req}$, °C days). In line with[15], $TU_{req}$ are derived consistently with the phenological module of the crop model LPJmL for each grid cell, crop, and rule-based computed growing period from the respective climate input. They are calculated as the sum of daily mean air temperature increments above a crop-specific base temperature (TU) (Supplementary Table 1) between rule-based sowing and maturity. In addition, winter-wheat cultivars require effective vernalization days ($VU_{req}$), that range between 0 (mild winters) and 70 (cold winters), depending on the

temperature of the 5 coldest months (Eq. (1))[15,18].

$$VU_{req} = \frac{70}{5} \times \left(1 - \frac{T_m - 3}{10 - 3}\right) \qquad (1)$$

where $T_m$ is the mean temperature of the month.

From the day of sowing, effective TU for phenological development are accumulated daily, as the difference between the mean air temperature on that day and the crop-specific base temperature for phenological development (Eq. (2)). The vernalization effectiveness is computed daily by a scaling factor (0–1), which is then multiplied to the TU (Eq. (2)). For crops that are insensitive to vernalization, $VU_d$ is set equal one.

$$TU_{req} = \sum_{d=1}^{ndays} \left( \max(0, T_d - T_{base}) \times \sum_{0}^{d} VU_d \right) \qquad (2)$$

where the scaling factor $VU_d$ is computed by a three-stage linear response function with a range of optimal temperatures (Eq. 3). Temperature for effective vernalization range between −4 °C and +17 °C, with an optimum range between 3 °C and 10 °C.

$$VU_d = \begin{cases} (T_d - (-4))/(3 - 10) & \text{if } -4 < T_d < 3 \\ 1 & \text{if } 3 \le T_d \le 10 \\ (17 - T_d)/(17 - 10) & \text{if } 10 < T_d < 17 \\ 0 & \text{otherwise} \end{cases} \qquad (3)$$

In this study, we have removed the effect of vernalization on slowing down TU accumulation until 10% of the total vernalization requirements is reached. In this way, the crop can accumulate both vernalization units and heat units in fall, so that there is some leaf growth before winter (in LPJmL, the LAI curve depends on accumulated heat units).

The LPJmL model simulates phenology as one single phase from emergence to maturity. Although the flowering stage is not simulated as an explicit break point, the fraction of above-ground biomass that is allocated to the storage organs (fHI) depends on the phenological progress (fTU$_{req}$, fraction of TU$_{req}$ that have been fulfilled), with the bulk of the storage organs start filling up after 40% of TU$_{req}$ have been reached (Eq. (4)). In line with this, the LAI curve reaches a plateau when 45% (wheat) or 50% (other crops) of the TU$_{req}$ are fulfilled, which could be considered a proxy of the flowering stage.

$$fHI = 100 \times \frac{fTU_{req}}{100 \times fTU_{req} + \exp^{11.1 - 10.0 \times fTU_{req}}} \qquad (4)$$

Crop biomass growth is simulated by daily carbon accumulation and allocation to different plant organs (roots, leaves, storage organs, mobile reserves, and stem). The fraction of carbon allocated to each pool is a function of the fraction of completed phenological progress. Water stress increases allocation to the roots and reduces allocation to the leaves. The daily Net Primary Production (NPP) is the result of the Gross Primary Production (daily gross photosynthesis) reduced by the respiration costs. Gross photosynthesis is simulated as a function of absorbed photosynthetically active radiation, $CO_2$ atmospheric mixing ratio, air temperature, day length, and canopy conductance. Photosynthesis rate is given by the minimum between light-limited and Rubisco-limited photosynthesis rates, with distinguished pathways for $C_3$ and $C_4$ crops. Respiration is tissue-specific and it is also driven by temperature. If accumulated NPP is insufficient to satisfy all organ demands, allocation follows a hierarchical order from roots, to leaves, to storage organs, and consequently penalizing the harvest index. Crops are subject to yield failure due to frost events (daily minimum temperature <−5 °C) occurring during grain yield formation (50% <fPHU <95%).

All management inputs other than sowing dates and cultivars have been provided according to global gridded datasets (0.5° × 0.5° spatial resolution) of current and historical practices. Land-use datasets for the period 1510–2015, of crop- and irrigation-specific area fraction and of mineral nitrogen fertilizer annual application rates (kg N ha$^{-1}$ year$^{-1}$), were based on the LUH2v2[55] dataset. This has been dis-aggregated and remapped using the MADRaT tool[56], in order to match the crop classification to the Crop Functional Types simulated by LPJmL[24]. The crop residue management input dataset (1850–2015) was obtained from ref. 56, which estimates crop residue removal rates from FAOSTAT national statistics on residue-related management practices (e.g., burning on field, household fuel and livestock fodder). The crop-specific spatial and temporal patterns of tillage and no-tillage systems for the period 1973–2010 were derived from ref. 57.

## Crop-model skills are preserved when driven by simulated crop calendars

We run LPJmL with both simulated and observed historical growing periods, forced by the historical observation-based climate dataset WFDEI[23], and test model outputs against a standard benchmark for global gridded crop models evaluation[58]. Forcing LPJmL with simulated crop calendars preserves the model skills in simulating global spatio-temporal patterns of crop yields, as indicated by the three metrics to compare simulated and observed national yields (correlation coefficient, the root mean squared difference, and the variance) (Supplementary Fig. 25).

## Data processing and analysis

We assess the future adaptation potentials over the present-day cropland[11]. To quantify the performances of cropping systems, we compute grid-cell level 20-years area-weighted average crop yields (t ha$^{-1}$) in each simulation period and GCM. We quantify climate change impacts with and without adaptation as the difference between future and historical yields. Furthermore, we isolate the adaptation effect from that of climate, by computing yield differences between contrasting management scenarios (e.g., with and without adaptation) under the same future period (2080–2099) for each climate scenario separately, where crops experience the same climate change scenario but are grown in different growing periods. As crop yield interannual variability metric, we use the coefficient of variation computed as year-to-year yield deviation normalized by the mean yield over the same period (1986–2005 for the reference scenario and 2080–2099 for all other scenarios). We account for uncertainties from climate scenarios by expressing the evaluation metrics at the aggregated level (e.g., global yield losses) as the mean and range across the four GCMs. In the main text, spatial patterns of growing period and yield changes are shown as the mean change per grid cell across four GCMs. The results for the individual GCMs are reported in the Supplementary Figures.

We compute the area-weighted mean absolute error (MAE) at the grid-cell and country levels as (Eq. (5)):

$$MAE = \frac{\sum_{i=1}^{n} |S_i - O_i| A_i}{\sum_{i=1}^{n} A_i} \qquad (5)$$

where $n$ is the number of grid cells (countries), $A_i$ is the crop-specific area of grid cell (country) $i$, $S_i$, and $O_i$ are the simulated and observed average dates (sowing or maturity) of grid cell (country) $i$ in Julian days.

For data processing, we used R[59] and R-packages for handling netcdf4[60], performing computation[61,62], and plotting results[63]. The country borders displayed in all figures that include maps were based on the Natural Earth dataset, which is available at naturalearthdata.com.

## Reporting summary

Further information on research design is available in the Nature Portfolio Reporting Summary linked to this article.

## Data availability

The raw outputs of the LPJmL crop model generated and used in this study have been deposited in the Zenodo database under accession code   https://doi.org/10.5281/zenodo.7038163   (https://zenodo.org/record/7038163#.YyRHObRBwaE). Climate reanalysis data (WFDEI) and model projection of the four GCMs (HadGEM2-ES, GFDL-ESM2M, IPSL-CM5A-LR, MIROC5) are available at https://www.isimip.org/. All other input data necessary to run the LPJmL model were obtained from external sources, as described in the Methods. They are archived in the internal PIK publication meta-database with the entry ID number 3369 and can be provided by the corresponding author to individual users who want to conduct the same analysis. Source data are provided with this paper.

## Code availability

The source code of the crop-calendar model, the source code of the LPJmL model, as well as the scripts workflow generated and used for the analysis have been deposited in the Zenodo database under accession code https://doi.org/10.5281/zenodo.7038163. An updated version of the crop-calendar model in form of an R package is made publicly available in the GitHub repository at https://github.com/AgMIP-GGCMI/cropCalendars.

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

## Acknowledgements

We acknowledge financial support from the MAPPY project (01LS1903A, S.M. and C.M.) funded through the German Federal Ministry of Education and Research (BMBF Germany). J.J. received funding from the Open Philanthropy Project and from the NASA GISS Climate Impacts Group. We acknowledge financial support from the International Wheat Yield Partnership (IWYP, grant IWYP115, S.A.). For A.U., this research was conducted under the USAID and Bill and Melinda Gates Foundation supported Cereal Systems Initiative for South Asia (CSISA; http://csisa.org/) and with the support of the CGIAR Regional Integrated Initiative Transforming Agrifood Systems in South Asia, or TAFSSA (https://www.cgiar.org/initiative/20-transforming-agrifood-systems-insouth-asia-tafssa/).

## Author contributions

Conceptualization: S.M., C.M., and J.J. Methodology: S.M., C.M., J.J., and A.U. Investigation: S.M. and J.J. Visualization: S.M. Writing—original draft: S.M. Writing—review & editing: S.M., J.J., S.A., A.U., and C.M.

## Funding

## Competing interests

The authors declare no competing interests.
