## [Peer Review File · Nature Communications]

Global crop yields can be lifted by timely adaptation of growing periods to climate changeReviewers' Comments:

Reviewer #1:

Remarks to the Author:

Review of "Integrated modelling of environmental and social benefits of land use policies in relation to Sustainable Development Goals"

The authors investigated the effects of adaptive management of crop phenology on global crop productivity (or production? see my remark below). This is an interesting, relevant subject and I enjoyed reading the manuscript. I have however the following major concerns about the manuscript:

1. In the introduction you rightly point out that phenology of plants is driven by temperature. However, I disagree with your statement that day length plays a role for only some species. To my understanding, phenological development of many important crops such as corn, wheat and potato is driven by temperature and day length. Day length is not considered in your methodology. It is not clear from your manuscript why you have not considered day length nor do you describe the implications of not considering it. This needs to be improved (either clearly and convincingly explaining why you have not considered it, including describing the implications; or to include day length effects in the simulations).
2. L.64-68: In addition to average future productivity, inter-annual variability is also important for food security, especially under climate change with higher expected inter-annual variability in weather conditions. Could you add this to your objectives/research questions and also show add these results?
3. L144: I appreciate your transparency on how daily weather is derived from monthly weather within LPMmL. However, it is also necessary to discuss the implications of this limitation (i.e. a likely overestimation of yields, see also: Spatial frameworks for robust estimation of yield gaps by Rattalino Edreira et al., 2021 in Nature food).
4. L188: These unexpected results may be due to methodology used. So I'd rather discuss this in the discussion section rather than assume these farmers aren't adapting (it also reads like a pretty bold assumption).
5. L266-269: I think this statement is too much: how can the presented work be the basis for these other decision making?
6. In general, I missed the social relevance in your manuscript. What are the current estimates of food production losses due to climate change? How does the 11% increase in global crop productivity from timely adaptation of sowing dates and cultivars compare to these estimates of food production losses due to climate change? And how does the 11% compare to the uncertainties in global crop model outcomes? In other words: the current manuscript is rather technical and could be made more interesting by including the societal relevance of the research.

Minor remarks:

- Abstract:

- Readers will probably not be able to understand the abstract without reading the full text. This needs to be adapted. For example, temperature-limited regions needs to be defined, as well as extra-tropics and what you mean with sizeable (last sentence).
- In addition, it is not clear from the abstract if crop productivity is referring to potential, water-limited or actual crop yields.
- I would also write that adaptive management of crop phenology is one of the central aspects, not the main one. Irrigation or changing crops are other examples of central aspects of crop production systems.
- In several sections you describe or show global crop productivity (e.g. Fig. 2), defined as yield x harvested area. Yield (t/ha) x harvested area (ha) is however considered production (t) in agronomy. Please correct this through the whole manuscript, because I think your definition is confusing.
- Check the figure caption of your supplementary material (some missing numbers and text is left).
- Fig. 1: Why did you display the differences between the two scenarios? Now it is not possible to see the how many days sowing dates were advanced or postponed.
- L46: the concept "maturity classes" is probably not known by main readers of Nature Communications

- L47: "it" to what do you refer?
- L58: to which "recent efforts" do you refer?
- L61-63: in the discussion you give examples of regional studies that investigated changing sowing dates and effects of crop productivity. I therefore think that your statement in L61-63 is a bit too negative. You could rephrase it to "remains poorly understood at the global scale."
- L64-68: what is your reference period, i.e. to which period do you compare sowing and harvest dates and crop productivity?
- L71: typo in "agro-climatic principles"
- L78-80: I don't understand the sentence: "We simulated 20-years average". What is meant in this respect with "counterfactual"? And to what did you match the sowing and harvest dates?
- L113-114: the sentence is confusing. I assume that these calculations crossed the year, it would be good to add that.
- L114: I don't think that "historic" is the best word in this context.
- L120: I don't think that diseases influence sowing dates a lot, other, more socio-economic circumstances such as labour will have a larger influence.
- L123-125: I think you draw the conclusion that the onset of the main rainy season is projected to not change much in the climate scenario too quickly. These results may also come out of your methodology.
- L168-178: why did you include simulations with elevated CO2 if this is such a large source of uncertainty in crop models?
- L260-263: The impact of climate change is indeed largest in vulnerable regions, but other factors than non-adapted cultivars to climate change play a larger role for determining yield levels (such as low fertilizer applications).
- L278: What is meant with "average growing seasons"? Average start and end? Or average length?
- L294-295: If temperature falls below the sowing temperature threshold before 15th of September in the Northern hemisphere, I would consider this a long winter, shouldn't it be after 15th of December?
- L318: I think you should also refer here to reference number 20.
- L375-381: Why did you consider mineral nitrogen fertilizer application rates (also check the period, I think there is a typo), crop residue management and tillage management? In the main text you describe that you have considered two water management regimes, but fertilizer, crop residue and tillage management is not mentioned, nor it is explained how these management influences crop biomass growth within LPJmL.
- L392-395: I don't understand from your text how you have isolated the adaptation effect from the climate change effect.

Reviewer #2:

Remarks to the Author:

The current study examines the integration of farmers' decision-making procedure with a process-based model at a global scale for wheat, maize, rice, soybean, and sorghum under climate change conditions, as an adaptation strategy. The authors predicted a possible shift in sowing dates of different study crops, particularly in temperature-limited areas. They also indicated the requirement for introducing late-maturity cultivars in extra-tropics. A combination of change in sowing date and modern cultivars would result in approximately 11% improvement in global crop productivity. This manuscript addresses an important topic as well as relevant research questions. A set of regional studies such as Huang et al 2020 (<https://doi.org/10.1088/1748-9326/ab66ca>) or Lv et al 2020 (<https://doi.org/10.1007/s11027-019-09861-w>) even at European scale (Parent et al 2018 -- <https://doi.org/10.1073/pnas.1720716115>) are also shown the relevance of a combination of change in sowing date and cultivar choice combating negative effects of climate change on crop productivity. The results are consistent with those sets of studies. It is undoubtedly a novelty in this research to consider the global level and compare different crops in this context. The article is very well written, and the figures are of brilliant quality. The methodology is well described so that the research is reproducible. Nevertheless, several conceptual and methodological issues need to be addressed before accepting the manuscript for publication as:

1- The manuscript relies on the adaptation of phenology but sowing and harvest which are at the

center of the study are false phenological phases (Estrella et al 2007 -- <https://doi.org/10.1111/j.1365-2486.2007.01374.x>). There is no information on the change in real phenological phases such as flowering and maturity within the text. More importantly, it is not clear how they switch between reproductive and grain filling phases. The modern cultivars generally reach flowering time faster than older cultivars and have a longer grain filling period. Therefore, determining the right time for the switch between phases is critical. In addition, flowering time is the most sensitive phenological stage to heat stress which increased under climate change conditions.

2- The authors also did not consider the effects of drought on crop phenology which is closely tied to temperature effects. When crops have enough water, they are less likely exposed to high temperatures since the transpiration cooling can reduce the crop temperature up to six degrees Celsius (Siebert et al 2014 -- <https://doi.org/10.1088/1748-9326/9/4/044012>). When drought stress occurs, crops close the stomata to preserve water means less transpiration. The reduction in transpiration can increase the crop surface temperature remarkably higher than the air temperature. Therefore, using air temperature for evaluating the effects of temperature increase on phenology advancement without considering canopy temperature (simplest way: establish a link between soil moisture and air temperature) can be misleading.

3- As far as I understood from the literature LPJmL uses a single base temperature from the whole growing period which is a strong simplification of natural conditions. It is well documented that the base temperatures for vegetative, reproductive, and grain filling phases are significantly different (Sánchez et al 2013 -- <https://doi.org/10.1111/gcb.12389> and Porter and Gawith 1999 -- [https://doi.org/10.1016/S1161-0301\(98\)00047-1](https://doi.org/10.1016/S1161-0301(98)00047-1)). Using multiple base temperatures can significantly change the results of the current study as it focused on phenology adaptation.

4- The authors switched off photoperiod response (line 346) in the simulations. Photoperiod has a strong impact on balancing the effects of temperature variability, particularly for winter wheat. Ignoring the photoperiod effect can result in an overestimation of temperature impact on crop phenology.

5- As the change in sowing date and maturity type can remarkably change the timing of the growing season it is not clear how the model deals with a possible change in intensity of frost damage and excessive water across different regions at the global scale.

Reviewer #3:

Remarks to the Author:

Farmers' adaptive management of crop phenology under climate change and their impact on global crop yield were poorly understood. An important reason is lacking sufficient information on farm management. This MS fills this gap by integrating models of farmers' decision-making with biophysical crop modelling (using LPJmL) at the global scale to simulate crop calendars adaptation (e.g. sowing date, harvesting date) to future climate (2080-2099, RCP6.0) and its effect on crop productivity.

The topic is important in the context of climate change in the search for food security options. The MS reads well and most of the conclusions are supported by the outcome of the modelling exercises. However, I have a few comments that deserve some attention.

- Reference scenario (or baseline) should be local cropping systems + local management practice (e.g. sowing date, harvesting date, rainfed/irrigated, etc.). It is not appropriate to assume that 'in each simulation unit different crops are assumed to grow on separate stands and to have only one crop cycle (sowing to-harvest) per year' and excluding the cropping systems like 'multi-cropping systems'. There were huge crop growing areas for these 'multi-cropping systems' which will have a profound effect/weight on the global grain production.
- Why did the authors choose the LPJmL model in this MS? It seems that the LPJmL model only considered the temperature effect on phenology. What about other factors such as the photoperiod effect on crop phenology?
- How did the authors calculate the global crop production? Is that crop area weighted yield? What are the weight factors for each crop at each country/grid cells? If yes, these values should be put

into supplementary tables.

- What is the heat and/or frost damage on crop production due to changed phenology under climate change?
- Fig 1. In middle part of China (North China Plain), the cropping systems is wheat and maize rotation, and maize was sown immediately after wheat harvest. This MS found for wheat the delayed sowing date and harvest date will be about 30-40 and 10-20 days, and for maize the sowing date will be about 30-40 days earlier. It seems impossible for an earlier maize sowing with a later wheat harvesting.

Specific comments/ suggestions:

Fig. S4: missing 'S4' in figure title.

Fig. S5: delete 'Type or paste caption here. Create a page break and paste in the Figure above the caption'.

Fig. S6: It is ok for India. But wheat in China should be separated into two parts (middle part and northeast part) in this study. Wheat in the northeast part of China should be spring wheat w/o vernal.

L111. 'Fig. Fig. 1A' should be 'Fig. 1A'.

L317. All winter type wheat cultivar requires the same effective vernalization days?

Revision of “Global crop yields lifted by timely adaptation of growing periods to climate change”

Sara Minoli, Jonas Jägermeyr, Senthold Asseng, Anton Urfels, Christoph Müller

Responses to the Reviewers

Reviewer #1 (Remarks to the Author):

Review of “Integrated modelling of environmental and social benefits of land use policies in relation to Sustainable Development Goals”

The authors investigated the effects of adaptive management of crop phenology on global crop productivity (or production? see my remark below). This is an interesting, relevant subject and I enjoyed reading the manuscript. I have however the following major concerns about the manuscript:

We thank the reviewer for his/her time to carefully review our manuscript and we highly appreciate the positive comment on the significance of the study. We address all critical points below.

1. In the introduction you rightly point out that phenology of plants is driven by temperature. However, I disagree with your statement that day length plays a role for only some species. To my understanding, phenological development of many important crops such as corn, wheat and potato is driven by temperature and day length. Day length is not considered in your methodology. It is not clear from your manuscript why you have not considered day length nor do you describe the implications of not considering it. This needs to be improved (either clearly and convincingly explaining why you have not considered it, including describing the implications; or to include day length effects in the simulations).

Thank you for raising this point. We agree that it is not correct to state that only in some species day length is a driver of phenology. What we intended to say was rather that, while all plant species respond to temperature, for some cultivated species day-facultative or day-neutral cultivars are available. For these cultivars, the sensitivity to photoperiod has been reduced or fully eliminated by breeding in order to allow adaptation of crops to a larger range of latitudes. Examples include day-neutral cultivars of maize grown in temperate regions (Hill & Li, 2006; Parent et al., 2018), rice grown in northern Asia (Hill & Li, 2006) and wheat grown in lower latitudes (van Bussel et al., 2015).

We have clarified the statement in **L46-52** as follows:

“Through plant domestication and breeding, humans have artificially modified the response of cultivated varieties (cultivars) to temperature and day length and by that, have expanded the area where crop species can be grown^{8,9}. Farmers can thus draw from a vast assortment of cultivars that differ by their thermal time (maturity classes), photoperiod (long-day, short-day or day-neutral) and vernalization requirements (vernalizing, non-vernalizing or facultative), which are considered central for adapting cropping systems to changing climatic conditions⁹.”

Moreover, we have added to the Materials & Methods section the explanation of the reason why we could not include photoperiod responses in our simulations (**L409-416**):

“All crops are assumed to be insensitive to photoperiod, due to a lack of parameters for multiple-crops and global-scale simulations. Previous global studies^{15,18} that have focused on maize and wheat only, have found lower performances in the growing period simulations when using a photo-thermal model, compared to a temperature-only driven approach and thus recommend caution when using the

photoperiodic response. State-of-the-art global crop models^{13,16} also typically do not consider sensitivity to photoperiod or assume that the photoperiodic response of the cultivars chosen in each location are perfectly tuned to the given conditions.”

Finally, we have discussed the implication of not including photoperiodic response in our simulations **(L307-315)**:

“Although LPJmL simulates phenology through a single phase from sowing to maturity, anthesis could be identified as the time when allocation to storage organs starts. For a correct parameterization and evaluation of such additional phase and to project changes in flowering and grain-filling time under future climate, global datasets on crop flowering dates would be needed. Particularly, time series of flowering dates would also help the improvement of the phenology model, including the effect of photoperiod. The omission of photoperiod responses in the current and previous global modelling assessment studies, might have led to an oversensitivity of crop growing periods to increasing temperature (see e.g.⁴⁹), which requires further investigation.”

2. L.64-68: In addition to average future productivity, inter-annual variability is also important for food security, especially under climate change with higher expected inter-annual variability in weather conditions. Could you add this to your objectives/research questions and also show add these results?

We agree that yield variability is an important aspect of food security that requires attention. We have extended our analysis to cover this by quantifying the coefficient of variation and its components (standard deviation and mean) of the annual yields within our reference and future periods of analysis. Results are shown in Fig. S16 and explained in **(L189-201)**:

“Another interesting aspect is the combined effects of increasing atmospheric CO₂, climate change and adaptation on yield variability, which we quantify by the coefficient of variation (Fig. S16). There is a general expectation that the interannual variability in crop yields will increase under climate change²⁸. Our simulations support this thesis only under the (unrealistic) scenario without CO₂ increase (no adaptation without CO₂), whereas we find even slight declines in yield variability if CO₂ is included (no adaptation). For rice sorghum and wheat, the two adaptation scenarios (timely adaptation and delayed adaptation) show a decreased variability compared to the no adaptation scenarios, because of negligible changes in standard deviation along with an increase in mean yield. On the other hand, yields of maize and soybean show increase in both mean and standard deviation, resulting in an overall higher coefficient of variation under adaptation scenarios, compared with no adaptation. However, also for these two crops neither the coefficient of variation nor the standard deviation of yields increase above those of the reference period.”

and in the methods section:

“As crop yield interannual variability metric, we use the coefficient of variation computed as year-to-year yield deviation normalized by the mean yield over the same period (1986-2005 for the reference scenario and 2080-2099 for all other scenarios).”

3. L144: I appreciate your transparency on how daily weather is derived from monthly weather within LPMmL. However, it is also necessary to discuss the implications of this limitation (i.e. a likely overestimation of yields, see also: Spatial frameworks for robust estimation of yield gaps by Rattalino Edreira et al., 2021 in Nature food).

Within LPJmL, we use daily weather data to compute phenological progress (i.e. accumulation of thermal units) and the primary productivity of the crops. Only the growing season algorithm is based on average monthly data to compute the seasonality of temperature and precipitation for each grid cell. Sowing date is computed if temperature trends (not individual days) cross the crop-specific thresholds (Waha et al. 2012, Minoli et al. 2019). Working with daily weather data would add little precision in computing temperature trends.

We recognize that this was not made clear enough in the manuscript and have improved the method description, clearly separating the mean growing periods calculation from the daily simulations into two separate sections:

- “Rule-based mean sowing and maturity dates”
- “Simulation of daily crop phenology and yields with the LPJmL crop model”

In **L155** we have pointed to the methods section for more details.

4. L188: These unexpected results may be due to methodology used. So I'd rather discuss this in the discussion section rather than assume these farmers aren't adapting (it also reads like a pretty bold assumption).

Indeed, farmers will very likely adapt to new climatic conditions, especially in regions with good market integration. We have decided to ignore adaptation if it leads to reduced productivity, simply because we only test one single adaptation measure here (changed growing seasons) and we cannot distinguish if the negative effect is driven by inadequate assessment of future growing seasons or by the inability of LPJmL to respond to the adverse conditions that the growing season algorithm tries to avoid. In practice, farmers will have several options to choose from (different sowing dates, different varieties, other management options) and changes to the system will be introduced gradually, with first movers trying new options and others following up, if the first movers are successful. However, if a change in growing seasons proves to be disadvantageous, the other farmers would not follow and first movers would try another combination of sowing dates and varieties. We don't explicitly represent the spread of new management practices and assume that all farmers adopt new growing seasons as computed by the model. Given that disadvantageous practices are unlikely to spread, we assume that in such cases, farmers would stay with the only other option they have in our model setup that is to stay with the old growing season.

We have added the explanation in **L211-218**:

“For each crop, a certain share of the cropland did not show an advantage for yields from adaptation (yields under timely adaptation \leq yields under no adaptation). In these cases, as we only test one single adaptation measure here (changed growing periods) and we cannot distinguish if the negative effect is driven by inadequate assessment of future growing seasons or by the inability of LPJmL to respond to the adverse conditions that the growing season algorithm tries to avoid, we assume that farmers would rather continue applying historical crop management. Given that disadvantageous practices are unlikely to spread, we assume that in such cases, farmers would stay with the only other option they have in our model setup that is to stay with the old growing period.”

5. L266-269: I think this statement is too much: how can the presented work be the basis for these other decision making?

We agree with the reviewer's comment on the too bold statement and have removed it.

6. In general, I missed the social relevance in your manuscript. What are the current estimates of food production losses due to climate change? How does the 11% increase in global crop productivity from timely adaptation of sowing dates and cultivars compare to these estimates of food production losses due to climate change? And how does the 11% compare to the uncertainties in global crop model outcomes? In other words: the current manuscript is rather technical and could be made more interesting by including the societal relevance of the research.

We acknowledge that the social relevance of the study was not made sufficiently clear in our manuscript. We have now put our results on the adaptation potentials into context, by reporting estimates of climate change impacts on crop yields from previous studies. We have also highlighted how adaptation is not only important for reducing the negative impacts of climate change, but it also enhances yields of the crops that benefit from increasing temperatures and CO₂. This is especially relevant in view of the growing population and consequent demand for food (L271-280).

“Climate change is expected to largely impact crop yields. The most recent global-scale estimates based on crop-model ensemble results (including LPJmL) show that, without accounting for adaptation under high-warming scenarios (RCP8.5), maize global yields might decline by -24%, wheat yields increase by +17%, whereas rice (+2%) and soybean (-2%) yields are only marginally affected¹³ (Table S3). Within that model ensemble, because of its strong response to atmospheric CO₂, LPJmL is on the more optimistic side, showing yield changes of -7%, +18%, +24%, +15% for maize, rice, soybean and wheat, respectively, which are in line with those found in our study for the same crops (-2%, +24%, +24%, +22%) (Fig. S14). Our finding that introducing adaptation might increase crop yields by 12%, demonstrates the importance of considering farmers’ decisions in global bio-physical modeling under climate change.”

Minor remarks:

- Abstract:

- Readers will probably not be able to understand the abstract without reading the full text. This needs to be adapted. For example, temperature-limited regions needs to be defined, as well as extra-tropics and what you mean with sizeable (last sentence).

- In addition, it is not clear from the abstract if crop productivity is referring to potential, water-limited or actual crop yields.

- I would also write that adaptive management of crop phenology is one of the central aspects, not the main one. Irrigation or changing crops are other examples of central aspects of crop production systems.

Thank you, we agree with your suggestions and have revised the abstract accordingly.

- In several sections you describe or show global crop productivity (e.g. Fig. 2), defined as yield x harvested area. Yield (t/ha) x harvested area (ha) is however considered production (t) in agronomy. Please correct this through the whole manuscript, because I think your definition is confusing.

The global yield of individual crops is computed as the crop area weighted yield. Since we consider the crop area to remain unchanged over time, the area weights are the same in the historical and future periods. Therefore, the relative difference in area-weighted yields (t/ha) are the same as those in production (t) (i.e. the numbers in Fig. 2 do not change if we report relative changes in production or in productivity).

From an agriculture economics perspective, production changes in the future will not depend on yield changes only, but also on land-use changes that are driven by changes in yields but by changes in socio-economic factors, too. It would not be correct to imply that if yields decline, global production will also decline, and vice versa. As we only model changes in yields (productivity) we prefer reporting changes in productivity, rather than production.

However, we agree that our previous definition generated confusion and the product of harvested area and yield is indeed the production (t) not the productivity (t/ha). We have corrected the statement in the captions of figures 2 and S14 and also replaced “productivity” with “yield” throughout the text, explaining that yield is area-weighted, where necessary.

- Check the figure caption of your supplementary material (some missing numbers and text is left).

Thanks, we have checked the supplementary materials.

- Fig. 1: Why did you display the differences between the two scenarios? Now it is not possible to see the how many days sowing dates were advanced or postponed.

We could not fully understand this comment, as in Fig. 1, we show the difference between the two scenarios (with adaptation, i.e. “timely adaptation” vs. without adaptation, i.e. “no adaptation”) in order to show by how many days adaptation shifts sowing and maturity dates. For example, in the Corn Belt (US), adaptation advances maize sowing dates by ~30 days, while it postpones wheat sowing dates by ~30 days.

- L46: the concept “maturity classes” is probably not known by main readers of Nature Communications

Along with the improved explanation on photoperiod response, we have clarified what we referred to by “maturity classes” (L49).

- L47: “it” to what do you refer?

As for the previous comment, we have rephrased the entire paragraph and addressed this comment as well.

- L58: to which “recent efforts” do you refer?

“Recent efforts” referred to the references 18-22 reported in the following sentence. We have now merged the two sentences and hope to have clarified it.

- L61-63: in the discussion you give examples of regional studies that investigated changing sowing dates and effects of crop productivity. I therefore think that your statement in L61-63 is a bit too negative. You could rephrase it to “remains poorly understood at the global scale.”

Thank you, we agree and have rephrased the sentence as suggested.

- L64-68: what is your reference period, i.e. to which period do you compare sowing and harvest dates and crop productivity?

The reference historical period in our study is 1986-2005, which we use to simulate historically adapted growing periods. The core of our analysis however, compares growing periods and yields in the future (2080-2099) under counterfactual adaptation scenarios. E.g. Fig 2 shows, yield differences between “adaptation” (“timely-adaptation”, “delayed-adaptation”, etc.) and “no-adaptation” scenarios in 2080-2099.

In order to clarify this as well as the comment on L78-80, we have rephrased the respective paragraphs. Changes can now be found in **L75-78**.

- L71: typo in “agro-climatic principles”

Typo has been corrected.

- L78-80: I don't understand the sentence: “We simulated 20-years average”. What is meant in this respect with “counterfactual”? And to what did you match the sowing and harvest dates?

We have addressed this point together with the one on L64-68. Please, refer to our response on that.

- L113-114: the sentence is confusing. I assume that these calculations crossed the year, it would be good to add that.

The rule-based computation of sowing and maturity does not look into a specific year, but returns average dates within an average climatic year. With the LPJmL model we have run parallel scenarios assuming adapted or non-adapted growing seasons. Therefore, within a crop-model run, we do not see such a switch between spring and winter types.

As suggested we have added a clarifying sentence (**L121-123**):

“Larger differences (>+60 days) found for wheat in high latitudes (>50°N-S) reflect the switch from spring to winter types, meaning that sowing dates are pushed several months forward from spring to fall.”

- L114: I don't think that “historic” is the best word in this context.

We have changed “historical” to “reference” and added the years 1986-2005 to highlight that this refers to what we called historical period, as opposed to the future 2080-2099 period.

- L120: I don't think that diseases influence sowing dates a lot, other, more socio-economic circumstances such as labour will have a larger influence.

Diseases might not be a big factor in sowing date decisions for spring crops. However, for winter wheat, which remains dormant over winter, the time of sowing is important because there needs to be a balance between a good emergence and crop establishment before entering winter, and avoiding excessive growth. The main reason why an excessive canopy development is not desirable is that larger plants are more susceptible to frost damage and to diseases (e.g. fungal pathogens) (Vico et al., 2014; Dueri et al., 2022).

- Vico, G., Hurry, V., & Weih, M. (2014). Snowed in for survival: Quantifying the risk of winter damage to overwintering field crops in northern temperate latitudes. *Agricultural and forest meteorology*, 197, 65-75.
- Dueri, S., Brown, H., Asseng, S., Ewert, F., Webber, H., George, M., ... & Martre, P. (2022). Simulation of winter wheat response to variable sowing dates and densities in a high-yielding environment. *Journal of Experimental Botany*.

- L123-125: I think you draw the conclusion that the onset of the main rainy season is projected to not change much in the climate scenario too quickly. These results may also come out of your methodology.

Thanks for this comment. We realized that there was an inconsistency in our methodology between the definition of the spring-temperature onset (for temperature-driven sowing dates) and the definition of the rainy-season onset (for precipitation-driven sowing dates). While the former was

based on daily (interpolated) values, the latter was defined based on monthly values (i.e. first day of the four wettest months), which did not allow to detect shifts in sowing dates smaller than 1 month. As you pointed out, this could have misled us to a wrong conclusion.

We have now modified the definition of the “main rainy season” to match the 120 consecutive wettest days in an average year. In this way, we can detect shifts in the order of days, as we do for temperature (see sowing date changes in tropical regions, Fig. 1).

We find that, even with this daily precipitation approach, the shifts in the main rainy season onset are relatively contained. We thus conclude that, on average, the timing of the rainy season does not change much, according to our approach. However, we discuss that future studies would need to investigate, not only average changes in seasonality, but also its interannual variability (L135-136):

“Yet, future approaches would need to address also changes in climate and growing periods interannual variability.”

• L168-178: why did you include simulations with elevated CO₂ if this is such a large source of uncertainty in crop models?

Greenhouse-gas emissions are the main climate forcing, driving the climate change scenarios generated by global circulation models, like the one we have used in this study (e.g. Taylor et al., 2012). Projecting climate change impacts on crop yield omitting the change in atmospheric CO₂ concentration, would neglect a central factor of the ongoing atmospheric changes as well as of crop photosynthetic activity. Despite remaining uncertainties in crop models CO₂ responses, climate change impact assessments should not omit scenarios with elevated CO₂. Some authors have even argued that scenarios without CO₂ increase should not even be reported, as they have generated more confusion than clarity and do not represent plausible future conditions (Toreti et al., 2020). Reporting the difference between model runs with and without CO₂ should not be interpreted as a measure of response-to-CO₂ uncertainty, but only be used to quantify the size of the CO₂ fertilization effect (Jägermeyr et al., 2021).

For this reason, our main scenario includes increasing CO₂. Yet, we decide to report results of a “without CO₂” scenario, to show that the assumptions on the CO₂ level does not affect the crop responses to growing period adaptation.

- Taylor, K. E., Stouffer, R. J., & Meehl, G. A. (2012). An overview of CMIP5 and the experiment design. *Bulletin of the American meteorological Society*, 93(4), 485-498.
- Toreti, A., Deryng, D., Tubiello, F. N., Müller, C., Kimball, B. A., Moser, G., ... & Rosenzweig, C. (2020). Narrowing uncertainties in the effects of elevated CO₂ on crops. *Nature Food*, 1(12), 775-782.
- Jägermeyr, J., Müller, C., Ruane, A. C., Elliott, J., Balkovic, J., Castillo, O., ... & Rosenzweig, C. (2021). Climate impacts on global agriculture emerge earlier in new generation of climate and crop models. *Nature Food*, 2(11), 873-885.

• L260-263: The impact of climate change is indeed largest in vulnerable regions, but other factors than non-adapted cultivars to climate change play a larger role for determining yield levels (such as low fertilizer applications).

We agree that there are other factors that might play a larger role in increasing crop yields than sowing dates and cultivars, such as fertilization and irrigation. In this paragraph we are just making the point that growing period adaptation is not a fully autonomous adaptation, especially in regard to cultivar changes. While shifting sowing dates can be directly implemented by farmers, without the need of

any additional inputs or technologies, cultivar change requires larger efforts that might constitute socio-economic barriers to implement adaptation in e.g. developing countries.

We have extended the sentence to cover this issue: *“As the impact of climate change in such more vulnerable regions is largest, concerted efforts are needed to exploit the full growing season adaptation potential, along with other measures to improve crop yields (e.g. improvement of soil fertility and water retention). These will depend on functional, fair, and sustainable markets in less developed regions, including the infrastructure necessary for accessing those markets⁴⁷” (L299-303).*

- L278: What is meant with “average growing seasons”? Average start and end? Or average length?

By “average growing seasons”, we meant the average start and end of the growing seasons. We have clarified this both in the first Material and Methods paragraph and in the section title “Rule-based mean sowing and maturity dates”.

- L294-295: If temperature falls below the sowing temperature threshold before 15th of September in the Northern hemisphere, I would consider this a long winter, shouldn't it be after 15th of December?

Thank you, this was a mistake in typing. If temperature falls below the sowing threshold before September 15th, then we consider winter as too long. The sentence has been corrected, taking into account an additional comment by Reviewer #3, as follows (L358-361):

*“For wheat we distinguish three types: winter wheat with vernalization is chosen if monthly temperatures fall below 0°C, but winter is neither too harsh (temperature of the coldest month is higher than -10°C), nor too long (temperatures fall below the sowing temperature threshold (12°C) **after** 15th September (North hemisphere) or 31st March (South hemisphere)¹⁹.”*

After 15th December, would instead be too late and it would exclude winter wheat from all those locations where this crop is sown e.g. in October or November, which is the most common case.

- L318: I think you should also refer here to reference number 20.

Reference added, thanks.

- L375-381: Why did you consider mineral nitrogen fertilizer application rates (also check the period, I think there is a typo), crop residue management and tillage management? In the main text you describe that you have considered two water management regimes, but fertilizer, crop residue and tillage management is not mentioned, nor it is explained how these management influences crop biomass growth within LPJmL.

We have considered nitrogen fertilizers, crop residues and tillage management to represent cropping systems as close as possible to current practices. This is a commonly used model set up for climate impact assessment studies, see for example the ISIMIP 3b protocol (<https://www.isimip.org/>). The intent is to quantify impacts on production systems assuming that no action would be taken in response to them. For our study, we use this assumption as the reference scenario and, on top of that, quantify the potential of growing period adaptation.

In the main text we have now specified that *“We use the process-based global gridded crop model LPJmL^{24,25} to simulate annual crop growing periods and yields from 1986 to 2099 under a reference scenario of actual crop- and grid-cell specific historical management practices (irrigation, fertilization, tillage and crop residue management), assuming that no management action would be taken in response to climate-induced impacts. We then compare the counterfactual growing periods (with*

adaptation) and corresponding yields at the end of the century (2080-2099) in order to quantify how much (i) future sowing and maturity dates would shift under different adaptation and climate scenarios and (ii) future crop yields could be enhanced, if farmers adapted growing periods to climate change.” (L81-88).

• L392-395: I don’t understand from your text how you have isolated the adaptation effect from the climate change effect.

In Fig. 2 and 3, we compare two adaptation settings (e.g. timely adaptation VS no adaptation) under the same climate time slice (2080-2099). In this way, we can isolate the adaptation effect on crop yields, as opposed to the combined climate and adaptation effects that we show in the Supplementary Information (Fig. S14).

We have explained the analysis in more detail, hoping to have clarified what we mean with “isolating the adaptation effect” (L479-484):

“We quantify climate change impacts with and without adaptation as the difference between future and historical yields. Furthermore, we isolate the adaptation effect from that of climate, by computing yield differences between contrasting management scenarios (e.g. with and without adaptation) under the same future period (2080-2099) for each climate scenario separately, where crops experience the same climate change scenario but are grown in different growing periods.”

Reviewer #2 (Remarks to the Author):

The current study examines the integration of farmers’ decision-making procedure with a process-based model at a global scale for wheat, maize, rice, soybean, and sorghum under climate change conditions, as an adaptation strategy. The authors predicted a possible shift in sowing dates of different study crops, particularly in temperature-limited areas. They also indicated the requirement for introducing late-maturity cultivars in extra-tropics. A combination of change in sowing date and modern cultivars would result in approximately 11% improvement in global crop productivity. This manuscript addresses an important topic as well as relevant research questions. A set of regional studies such as Huang et al 2020 (<https://doi.org/10.1088/1748-9326/ab66ca>) or Lv et al 2020 (<https://doi.org/10.1007/s11027-019-09861-w>) even at European scale (Parent et al 2018 -- <https://doi.org/10.1073/pnas.1720716115>) are also shown the relevance of a combination of change in sowing date and cultivar choice combating negative effects of climate change on crop productivity. The results are consistent with those sets of studies. It is undoubtedly a novelty in this research to consider the global level and compare different crops in this context. The article is very well written, and the figures are of brilliant quality. The methodology is well described so that the research is reproducible. Nevertheless, several conceptual and methodological issues need to be addressed before accepting the manuscript for publication as:

We thank the reviewer for his/her time to review our manuscript. We greatly appreciate all the positive comments, and particularly the recognition of the novelty of the global-scale approach in simulating adaptation. We have carefully considered all critical points and addressed them as explained below.

1- The manuscript relies on the adaptation of phenology but sowing and harvest which are at the center of the study are false phenological phases (Estrella et al 2007 -- <https://doi.org/10.1111/j.1365-2486.2007.01374.x>). There is no information on the change in real phenological phases such as flowering and maturity within the text. More importantly, it is not clear how they switch between

reproductive and grain filling phases. The modern cultivars generally reach flowering time faster than older cultivars and have a longer grain filling period. Therefore, determining the right time for the switch between phases is critical. In addition, flowering time is the most sensitive phenological stage to heat stress which increases under climate change conditions.

Thanks for pointing out these important issues.

It is true that sowing and harvest are false phenological phases and are rather farming management practices. In global-scale models sowing and harvest are often used interchangeably with emergence and maturity, respectively. This is due to lack of global-scale datasets on actual phenological phases and spatial patterns of crop-phenology parameters, such as growing degree days or sensitivity to photoperiod. Therefore, reported dates of sowing and harvest are used as proxies for emergence and maturity dates for calibrating growing periods for global model experiments. Moreover, global gridded crop models often simulate phenology as one single phase from emergence to maturity and the flowering stage is either not explicitly simulated or not calibrated against observational datasets (e.g. Minoli et al., 2019 - DOI: 10.1029/2018EF001130). Yet, we agree that a clearer definition of the terminology is necessary and modified the text accordingly:

- In the introduction, we have now clarified that there are different aspects limiting the study of crop phenology at the global scale, by modifying former L57-58 to: *“Studying changes in global crop phenology is constrained by the lack of sufficient information on farm growing-period management (e.g. sowing and harvest dates, cultivar choice), timing of phenological phases (e.g. flowering, maturity) and crop development parameters (e.g. growing degree days, base temperatures, sensitivity to photoperiod).”*
- Due to the several limitations in understanding global phenology, in this study, we are bound to investigate farm growing-period management only. Therefore, we have now modified the title to *“Global crop productivity lifted by timely adaptation of growing periods to climate change”*. Moreover, we have replaced *“phenology”* with *“growing periods”* throughout the text, where appropriate.
- The LPJmL model simulates phenology as one single phase from emergence to maturity and the flowering stage is not simulated as an explicit break point. However, the fraction of above ground biomass (AGB) that is allocated to the storage organs depends on the phenological progress (fraction of PHU requirements that have been fulfilled). As you can see from Fig. R1, the bulk of the storage organs start filling up after 40% of Phenological Heat Unit (PHU) requirements have been reached. In line with this, the LAI curve reaches a plateau at 45% (wheat) or 50% (other crops) of the PHU requirements are fulfilled, which could be considered a proxy of the flowering stage.
- *Figure R1. harvest index (optimal) as a function of the fraction of total PHU requirements (fPHU).*

- In the rule-based growing period approach, it is assumed that the reproductive phase has a static duration of 60 days for maize and 40 days for all the other crops. This assumption is based on empirical relationships between total growth period and reproductive phase, showing that the partition between vegetative and reproductive phase of grain crops follows a saturation curve (Egli, 2011), that levels off after 90-100 days of total growth duration (Fig. R2). As we do not simulate growing periods shorter than 90 days, the reproductive period is assumed to be static. Moreover, flowering dates are not an output of the crop calendar model, which only returns sowing and harvest dates, but are only internally used in the model for estimating the average timing of the grain-filling phase.

- Fig. R2 The relationship behind the reproductive phase static-length assumption (Egli, 2011).

- We acknowledge that the limitations reported above are critical, but argue that such limitations existed also in all previous global impact assessments carried out with process-based crop models so far. With this paper we tackle one of several limitations in global crop yield projections, which is the adaptation of growing periods through management of sowing dates and cultivars. All other aspects will need to be addressed in future research, but are out of scope for our study.

- We improved the description of the methodology and discussed its limitations:

- **L375-382:** “Note that the grain-filling phase has a static duration of 60 days for maize and 40 days for all the other crops. This assumption is based on empirical relationships between total growth period and the post-flowering reproductive phase, showing that the partition between vegetative and reproductive phase of grain crops follows a

saturation curve that levels off after 90-100 days of total growth duration⁴⁹. Detailed description of the rules and parameterization can be found in ^{19,22}.”

- **L439-445:** “The LPJmL model simulates phenology as one single phase from emergence to maturity. Although the flowering stage is not simulated as an explicit break point, the fraction of above ground biomass that is allocated to the storage organs depends on the phenological progress (fTU_{req} , fraction of TU_{req} that have been fulfilled), with the bulk of the storage organs start filling up after 40% of TU_{req} have been reached (Eq. 4). In line with this, the LAI curve reaches a plateau when 45% (wheat) or 50% (other crops) of the TU_{req} are fulfilled, which could be considered a proxy of the flowering stage.”
- **L305-318:** “We identify two limitations that LPJmL has in common with several other state-of-the-art global-scale crop models (e.g. ⁴⁸). First, the adaptation of sowing dates and cultivars will not only affect the start and end of growing periods, but also the timing of intermediate phenological phases, some of which are particularly susceptible to high temperatures (e.g. flowering). Although LPJmL simulates phenology through a single phase from sowing to maturity, anthesis could be identified as the time when allocation to storage organs starts. For a correct parameterization and evaluation of such additional phase and to project changes in flowering and grain-filling time under future climate, global datasets on crop flowering dates would be needed. Particularly, time series of flowering dates would also help the improvement of the phenology model, including the effect of photoperiod. The omission of photoperiod responses in the current and previous global modelling assessment studies, might have led to an oversensitivity of crop growing periods to increasing temperature (see e.g. ⁴⁹), which requires further investigation. Including intermediate phenological phases would also allow to test the sensitivity of the model to the use of phase-specific base temperatures⁵⁰, instead of a single base temperature as assumed by most global scale models so far¹⁷.”

2- The authors also did not consider the effects of drought on crop phenology which is closely tight to temperature effects. When crops have enough water, they are less likely exposed to high temperatures since the transpiration cooling can reduce the crop temperature up to six degrees Celsius (Siebert et al 2014 -- <https://doi.org/10.1088/1748-9326/9/4/044012>). When drought stress occurs, crops close the stomata to preserve water means less transpiration. The reduction in transpiration can increase the crop surface temperature remarkably higher than the air temperature. Therefore, using air temperature for evaluating the effects of temperature increase on phenology advancement without considering canopy temperature (simplest way: establish a link between soil moisture and air temperature) can be misleading.

We recognize that using air temperature instead of canopy temperature might introduce some biases in the simulation of crop phenological progress and other plant processes, particularly under water stress conditions. However, this is a limitation that exists in the vast majority of process-based crop models, both at the field (Rezaei et al., 2015) and the global scale (Minoli et al., 2019). The challenges in using canopy temperature are several, going from the less standardized measurements of canopy temperature in the field to the data- and time-expensive calibration and testing procedures of the models (Webber et al., 2016).

We have included a paragraph in the discussion, to explain the limitations of the LPJmL model in this regard and suggested ways forward for future research:

L318-327: *“Second, LPJmL is a global crop model that is restricted in its representation of extreme-climate impacts on crop phenology and growth, such as heat stress, frost damage and soil water excess. Improving the representation of these impacts will be necessary to increase confidence in the estimated climate change impacts and potentials of growing periods adaptation on global food yields. For example, using air temperature instead of canopy temperature might introduce some biases in the simulation of crop phenological progress and other plant processes, particularly under water stress conditions. On the other hand, the modelling of canopy temperature is still subject to several challenges (e.g. less standardized measurements of canopy temperature in the field to the data- and time-expensive calibration) and has not been tested globally⁵¹, which makes it a longer-term objective for model development.”*

- Rezaei et al., 2015 - <http://dx.doi.org/10.1016/j.eja.2014.10.003>
- Minoli et al., 2019 - <https://doi.org/10.1029/2018EF001130>
- Webber et al., 2016 - <https://doi.org/10.1016/j.envsoft.2015.12.003>

3- As far as I understood from the literature LPJmL uses a single base temperature from the whole growing period which is a strong simplification of natural conditions. It is well documented that the base temperatures for vegetative, reproductive, and grain filling phases are significantly different (Sánchez et al 2013 -- <https://doi.org/10.1111/gcb.12389> and Porter and Gawith 1999 -- [https://doi.org/10.1016/S1161-0301\(98\)00047-1](https://doi.org/10.1016/S1161-0301(98)00047-1)). Using multiple base temperatures can significantly change the results of the current study as it focused on phenology adaptation.

We agree with the reviewer’s comment. The phenology module in LPJmL is relatively simple, as it simulates phenology as one single phase from emergence to maturity. It has been developed following the EPIC model in order to have a robust approach, especially for global-scale applications, where the calibration of phenology parameters is particularly challenging and relies on uncertain datasets. For this reason, we focus on the full growing period duration, rather than on individual stages. We discuss possible implications of this simplification, but argue that simulations may not be as sensitive to specifically parameterizing the base temperature of the reproductive and grain filling period as these periods are typically warm(er) and actual temperatures rarely are at the base temperature relevant for these phases.

We have added the following statement (**L315-318**):

“Including intermediate phenological phases would also allow to test the sensitivity of the model to the use of phase-specific base temperatures⁵⁰, instead of a single base temperature as assumed by most global scale models so far¹⁷.”

4- The authors switched off photoperiod response (line 346) in the simulations. Photoperiod has a strong impact on balancing the effects of temperature variability, particularly for winter wheat. Ignoring the photoperiod effect can result in an overestimation of temperature impact on crop phenology.

We have added to the Methods section the explanation of the the reason why we could not include photoperiod responses in our simulations (**L409-416**):

“All crops are assumed to be insensitive to photoperiod, due to a lack of parameters for multiple-crops and global-scale simulations. Previous global studies^{15,18} that have focused on maize and wheat only, have found lower performances in the growing period simulations when using a photo-thermal model, compared to a temperature-only driven approach and thus recommend caution when using the photoperiodic response. State-of-the-art global crop models^{13,16} also typically do not consider

sensitivity to photoperiod or assume that the photoperiodic response of the cultivars chosen in each location are perfectly tuned to the given conditions.”

We have discussed the implication of not including photoperiodic response in our simulations (**L307-315**):

“Although LPJmL simulates phenology through a single phase from sowing to maturity, anthesis could be identified as the time when allocation to storage organs starts. For a correct parameterization and evaluation of such additional phase and to project changes in flowering and grain-filling time under future climate, global datasets on crop flowering dates would be needed. Particularly, time series of flowering dates would also help the improvement of the phenology model, including the effect of photoperiod. The omission of photoperiod responses in the current and previous global modelling assessment studies, might have led to an oversensitivity of crop growing periods to increasing temperature (see e.g.⁴⁹), which requires further investigation.”

Please, refer to our response to reviewer #1 point 1 for further details.

5- As the change in sowing date and maturity type can remarkably change the timing of the growing season it is not clear how the model deals with a possible change in intensity of frost damage and excessive water across different regions at the global scale.

Excessive water is not well represented by most global gridded crop models (e.g. Li et al. 2019), as is the case in LPJmL. Frost damage is explicitly considered, although in a simplified way, and would thus also affect crop growth in all (adapted or non-adapted) growing seasons.

We specified this in the methods section:

“Crops are subject to yield failure due to frost events (daily minimum temperature < -5°C) occurring during grain yield formation (50% < fPHU < 95%).”

- Li, Y., Guan, K., Schnitkey, G. D., DeLucia, E., & Peng, B. (2019). Excessive rainfall leads to maize yield loss of a comparable magnitude to extreme drought in the United States. *Global change biology*, 25(7), 2325-2337.

Reviewer #3 (Remarks to the Author):

Farmers' adaptative management of crop phenology under climate change and their impact on global crop yield were poorly understood. An important reason is lacking sufficient information on farm management. This MS fills this gap by integrating models of farmers' decision-making with biophysical crop modelling (using LPJmL) at the global scale to simulate crop calendars adaptation (e.g. sowing date, harvesting date) to future climate (2080-2099, RCP6.0) and its effect on crop productivity.

The topic is important in the context of climate change in the search for food security options. The MS reads well and most of the conclusions are supported by the outcome of the modelling exercises. However, I have a few comments that deserve some attention.

We thank the reviewer for his/her time to review our manuscript and for acknowledging the importance of the topic. We greatly appreciate all the positive comments and have carefully considered all critical points, addressing them as explained below.

- Reference scenario (or baseline) should be local cropping systems + local management practice (e.g. sowing date, harvesting date, rainfed/irrigated, etc.). It is not appropriate to assume that 'in each simulation unit different crops are assumed to grow on separate stands and to have only one crop

cycle (sowing to-harvest) per year' and excluding the cropping systems like 'multi-cropping systems'. There were huge crop growing areas for these 'multi-cropping systems' which will have a profound effect/weight on the global grain production.

Our reference scenario is indeed our best possible representation of local land use and management practices. As stated in the methods, all management inputs other than sowing dates and cultivars have been provided according to global datasets of current and historical practices. These include, land-use area per crop and irrigation system, fertilizers application rates, tillage system and crop residue management. We use simulated sowing and maturity dates also in both future and reference scenarios in order to make the two directly comparable. We demonstrate that the simulated sowing and maturity dates are in line with observations and thus represent well the historical practices.

Yet, we agree with the reviewer's comment that agricultural systems are much more complex than what is represented in the LPJmL model. In global crop modelling, the implementation of multi cropping systems and crop rotations is still in a very early stage, and the rule-based approach to simulate sowing and maturity dates has been designed to cover the main growing season of major grain crops.

Although different crops are indeed grown on separate stands, and thus their growing seasons are independent of each other, we do account for the crop rotation effects on main soil properties:

- 1) The size of each stand within a simulation unit (grid-cell) is defined by the input land-use dataset.
- 2) During a crop growing season, each stand is treated separately in terms of soil water, soil carbon and management. Yet, after harvest, the crop stand is merged into a set-aside stand and the soil properties of the two stands are weighted by area and averaged.
- 3) At the day of sowing of the following crop, the necessary land is taken from the set-aside stand to grow this crop.

This allows for accounting for the carry-over effects between different crops and years.

- Why did the authors choose the LPJmL model in this MS? It seems that the LPJmL model only considered the temperature effect on phenology. What about other factors such as the photoperiod effect on crop phenology?

We chose the LPJmL model, simply because we are part of the developing team of this model and have direct access to the source code and computing resources. The omission of photoperiod is not necessarily a problem of the model, which had this functionality implemented in the original model description (Bondeau et al. 2007). However, photoperiodic sensitivity is difficult to parameterize at the global scale and phenological models tested for global-scale application show better skill without considering photoperiod (van Bussel et al. 2018). Please, see also the responses to reviewers 1 and 2.

- How did the authors calculate the global crop production? Is that crop area weighted yield? What are the weight factors for each crop at each country/grid cells? If yes, these values should be put into supplementary tables.

The yield of individual crops is computed as the crop area weighted yield, meaning that the total sum of grid-cell level production (crop-specific yield gain multiplied by harvested area) divided by the sum of crop-specific harvested area. The grid-cell and crop specific area (ha) is obtained by the MIRCA2000 dataset (Portmann et al., 2010) and it is kept static over time. We have now plotted the crop-area maps separately for rainfed and irrigated crops (Fig. S13).

- What is the heat and/or frost damage on crop production due to changed phenology under climate change?

All yield limiting factors considered by the model (high temperature, frost, water limitations, nitrogen limitations) are explicitly accounted for in the simulations. We did not analyze the individual effects by turning effects off individually, as this is not the objective of our study and would require substantial computing resources.

- Fig 1. In middle part of China (North China Plain), the cropping systems is wheat and maize rotation, and maize was sown immediately after wheat harvest. This MS found for wheat the delayed sowing date and harvest date will be about 30-40 and 10-20 days, and for maize the sowing date will be about 30-40 days earlier. It seems impossible for an earlier maize sowing with a later wheat harvesting.

We acknowledge that not representing multi-cropping systems is a limitation of our study. Moreover, we focus on grain production only and we do not account for crops that are cultivated as forage, e.g. silage maize. To our knowledge, in the North China Plain, maize in rotation with winter wheat is grown as “summer maize” (e.g. Luo et al., 2020), which is mostly used for silage production for dairy farms (Liang et al., 2020).

We have added this limitation in the discussion section of the manuscript:

“Yet, some adaptation decisions remain excluded from our approach, which does not consider multiple-cropping systems nor crop rotations. In these systems, the end of the first growing season often determines the beginning of the second, adding complexity that is not captured by our climate-only driven approach.”

- Luo, Y., Zhang, Z., Chen, Y., Li, Z., & Tao, F. (2020). ChinaCropPhen1km: a high-resolution crop phenological dataset for three staple crops in China during 2000–2015 based on leaf area index (LAI) products. *Earth System Science Data*, 12(1), 197-214.
- Liang, S., Zhang, X., Lu, Y., An, P., Yan, Z., & Chen, S. (2020). Performance of double cropping silage maize with plastic mulch in the North China Plain. *Agronomy Journal*, 112(5), 4133-4146.

Specific comments/ suggestions:

Fig. S4: missing ‘S4’ in figure title.

Corrected.

Fig. S5: delete ‘Type or paste caption here. Create a page break and paste in the Figure above the caption’.

Deleted.

Fig. S6: It is ok for India. But wheat in China should be separated into two parts (middle part and northeast part) in this study. Wheat in the northeast part of China should be spring wheat w/o vernal.

Thank you for spotting this. Wheat in North-East China should indeed be simulated as spring wheat as also shown by Reynolds and Braun (2019). We have improved our decision rule for deciding whether to plant spring or winter wheat by adding one condition based on the temperature of the coldest month (**L358-361**):

“For wheat we distinguish three types: winter wheat with vernalization is chosen if monthly temperatures fall below 0°C, but winter is neither too harsh (temperature of the coldest month is higher

than -10°C), nor too long (temperatures fall below the sowing temperature threshold (12°C) after 15th September (North hemisphere) or 31st March (South hemisphere)¹⁹.”

The resulting map (“New version” in the figure below) shows a better agreement with spring and winter wheat patterns at higher latitudes.

L111. ‘Fig. Fig. 1A’ should be ‘Fig. 1A’.

We have corrected that.

L317. All winter type wheat cultivar requires the same effective vernalization days?

No, the vernalization days required vary depending on the temperatures of the five coldest months and are computed with Eq. 1. We have rephrased the sentence (L423-425), hoping to have made it clearer:

“In addition, winter-wheat cultivars require effective vernalization days (VU_{req}), that range between 0 (mild winters) and 70 (cold winters), depending on the temperature of the five coldest months (Eq. 1)^{15,18}.”

Reviewers' Comments:

Reviewer #1:

None

Reviewer #2:

Remarks to the Author:

The authors responded clearly and professionally to all of my comments and concerns. I enjoyed reading the manuscript and have no further comments.

Reviewer #3:

Remarks to the Author:

The authors have well understood my concerns and have provided convincing answers to all of them through additional data analysis and/or paragraph clarification.